# Circulating Natural Killer Cells as Prognostic Value for Non-Small-Cell Lung Cancer Patients Treated with Immune Checkpoint Inhibitors: Correlation with Sarcopenia

**DOI:** 10.3390/cancers15143592

**Published:** 2023-07-12

**Authors:** Marta Tenuta, Carla Pandozzi, Francesca Sciarra, Federica Campolo, Alain J. Gelibter, Grazia Sirgiovanni, Enrico Cortesi, Andrea Lenzi, Andrea M. Isidori, Emilia Sbardella, Mary Anna Venneri

**Affiliations:** 1Department of Experimental Medicine, Sapienza University of Rome, Viale Regina Elena 324, 00161 Rome, Italy; marta.tenuta@uniroma1.it (M.T.); carla.pandozzi@uniroma1.it (C.P.); francesca.sciarra@uniroma1.it (F.S.); federica.campolo@uniroma1.it (F.C.); andrea.lenzi@uniroma1.it (A.L.); andrea.isidori@uniroma1.it (A.M.I.); 2Medical Oncology Unit B, Policlinico Umberto I, Sapienza University of Rome, 00185 Rome, Italy; alain.gelibter@uniroma1.it (A.J.G.); enrico.cortesi@uniroma1.it (E.C.); 3Medical and Translational Oncology, Oncology Department, AO Santa Maria, 05100 Terni, Italy; g.sirgiovanni@aospterni.it

**Keywords:** immune checkpoint inhibitors (ICIs), natural killer (NK) cells, sarcopenia, peripheral blood mononuclear cells (PBMCs), lung cancer, non-small-cell lung cancer (NSCLC)

## Abstract

**Simple Summary:**

In this study, we provide a basal and longitudinal evaluation of immune cells in advanced non-small-cell lung cancer (NSCLC) patients undergoing PD-1 or PD-L1 blockade. We aimed to explore if any data could be predictive of better outcomes and long-term survival and to detect eventual connections among immune cell subsets and sarcopenia, another known risk factor for progression disease (PD). We found that natural killer (NK) cell basal levels are higher in patients with disease control (DC) compared to PD patients; higher NK cell basal levels predict a longer overall survival (OS); lower NK values represent a risk factor for PD; and after three months of immune check-point inhibitors (ICIs) treatment, NK cells (and the subclass CD56^bright^) significantly increase in DC patients. Interestingly, sarcopenic patients show lower NK cell values at basal levels.

**Abstract:**

Background: Immune checkpoint inhibitors (ICIs) have revolutionized the treatment of tumors. Natural killer (NK) cells can play an important role in cancer immune surveillance. The aim of this prospective observational study was to analyze peripheral blood mononuclear cells (PBMCs) in patients with advanced non-small-cell lung cancer (NSCLC) receiving ICIs in order to identify predictive factors for better survival outcomes. Methods: Forty-seven stage IV NSCLC patients were enrolled. Patients underwent baseline (T_0_) and longitudinal (T_1_) evaluations after ICIs. Peripheral immune blood cell counts were analyzed using flow cytometry. Results: Basal levels of CD3^−^CD56^+^ NK cells were higher in patients with controlled disease (DC) compared to progression disease (PD) patients (127 cells/µL vs. 27.8 cells/µL, *p* < 0.001). Lower NK cell values were independent prognostic factors for shorter overall survival (OS) (HR 0.992; 95% CI 0.987–0.997, *p* < 0.001) and progression-free survival (PFS) (HR 0.988; 95% CI 0.981–0.994, *p* < 0.001). During the longitudinal evaluation, CD3^−^CD56^+^ NK cells (138.1 cells/µL vs. 127 cells/µL, *p* = 0.025) and CD56^bright^ NK cells (27.4 cells/µL vs. 18.1 cells/µL, *p* = 0.034) significantly increased in the DC group. Finally, lower values of CD3^−^CD56^+^ NK cells (28.3 cells/µL vs. 114.6 cells/µL, *p* = 0.004) and CD56^dim^ NK cells (13.2 cells/µL vs. 89.4 cells/µL, *p* < 0.001) were found in sarcopenic patients compared to patients without sarcopenia. Conclusions: Peripheral NK cells could represent a non-invasive and useful tool to predict ICI therapy response in NSCLC patients, and the association of low NK cell levels with sarcopenia deserves even more attention in clinical evaluation.

## 1. Introduction

In the past 10 years, the clinical application of immune checkpoint inhibitors (ICIs) has significantly improved the prognosis of patients with advanced solid tumors by targeting immune inhibitory pathways that cancer cells frequently exploit to avoid detection and regulate immune proliferation and survival [1,2,3,4,5].

These drugs are monoclonal antibodies targeting some specific molecular checkpoints present on the surface of various immune cells able to prevent immune cells from functioning, therefore activating immune recognition and the consequent destruction of cancer cells [6,7]. There is a significant heterogeneity in clinical response, with patients experiencing no objective advantages from therapy; therefore, it is crucial to identify which patients would benefit from this specific treatment [8]. The prediction of ICIs’ benefits remains a critical unsolved challenge to increasing the overall efficacy rate and/or reducing unnecessary overtreatment.

Numerous studies have investigated the role of tumor-associated immune cell phenotypes related to immunotherapy outcomes [9,10]. CD4^+^ and CD8^+^ T lymphocytes play a primary role in the anti-tumor immunity activated by ICIs, but even non-T cell populations can be important players. Specifically, natural killer (NK) cells are a subset of innate lymphoid cells capable of killing tumor cells directly without antigen presentation [11,12,13]. Therefore, they are able to escape some of the cancer cell strategies to bypass the immune system, such as the downregulation (or complete loss) of human leukocyte antigen (HLA-I) molecules [14]. Peripheral blood is easily accessible for serial analysis compared to tumor biopsies, and it can serve as a surrogate measurement of the tumor’s interaction with host immune cells.

Sarcopenia has long been associated with higher toxicity induced by anti-cancer treatments and shorter survival in patients with solid tumors [15]. At the same time, sarcopenia is associated with high levels of inflammatory markers and cytokines; a pro-inflammatory status is also found in cancer patients with a worse prognosis. On this basis, sarcopenia seems to reflect the increased metabolic activity of more aggressive tumors, which involves systemic inflammation and muscle wasting and may have a negative role in the therapeutic response to ICIs. In this specific scenario, the literature is still scanty and limited to a few retrospective reports.

The main objective of this study was to analyze peripheral blood mononuclear cells (PBMCs) in a cohort of patients with advanced non-small-cell lung cancer (NSCLC) receiving anti-PD-1/PD-L1 in order to identify biomarkers that reflect cancer–immune cell interactions and could reliably reveal patient responsiveness to immunotherapy and predict better survival. Secondary objectives included the assessment of inflammatory markers, a longitudinal evaluation of PBMCs (after 3 months of ICI treatment), and PBMC assessment in the context of sarcopenia, a known risk factor for progression disease (PD) and a worse survival outcome [15].

We indeed hypothesize the association of sarcopenia with circulating immune cell phenotype dynamics in NSCLC patients treated with ICIs, therefore contributing to influencing their survival and response to treatment.

## 2. Materials and Methods

### 2.1. Study Design and Population

The population and study design, inclusion and exclusion criteria, and procedures have been extensively reported elsewhere [15]. Briefly, this is a proof-of-concept prospective longitudinal observational study. The study population included patients entering Oncology Unit B, Policlinico Umberto I of Rome, from October 2017 to February 2020 with a diagnosis of advanced NSCLC who were candidates to start anti-PD-1 or anti-PD-L1 (nivolumab, pembrolizumab, atezolizumab). The histological evaluation was performed by specialists in pathology trained in PD-L1 evaluation with PD-L1 SP263, a rabbit monoclonal antibody (Ventana Medical Systems Inc., Tucson, AZ, USA). The expression of PD-L1 was evaluated on all tumor cells (minimum of 200 neoplastic cells in each biopsy sample). A positive stain was defined as the presence of membrane staining, either strong or weak, complete or incomplete, in a percentage of cells ≥ 1%.

At the time of recruitment, ICIs had only been approved as monotherapy for NSCLC in Italy. Clinical and survival evaluations were carried out until 31 March 2023.

Differently from our previous paper [15], here we also included a longitudinal evaluation of the patients: in addition to baseline assessment (T_0_), a second evaluation was performed 3 months (±2 weeks) after the first ICI administration (T_1_), corresponding to the first radiological evaluation. T_0_ evaluation included a complete medical visit with assessment of performance status according to the Eastern Cooperative Oncology Group (ECOG) scale, morning blood sampling after overnight fasting, and Dual-energy X-ray absorptiometry (DXA). T_1_ evaluation included a complete medical visit with assessment of clinical response to treatment and a morning blood sampling after overnight fasting.

This study was approved by the Ethics Committee of Policlinico Umberto I (Ref. CE 4946) and was conducted in accordance with the Declaration of Helsinki principles with the prior written informed consent of each patient to participate in the study.

### 2.2. Blood Sample Processing and Immunophenotyping by Flow Cytometry

Blood samples were collected from patients and controls at 8:00 a.m. after overnight fasting. Routine full blood counts and inflammatory markers (erythrocyte sedimentation rate, ESR; C-reactive protein, CRP; fibrinogen; ferritin; transferrin) were performed. Absolute cell counts were derived from the total blood cell counts provided by the hematological analyzers (SYSMEX Roche, Indianapolis, IN, USA).

PBMCs were isolated from fresh whole blood using a Ficoll–Paque density gradient for cytometry analyses and gated as detailed in Figure 1. Fc-blocked PBMCs were stained with the following monoclonal antibodies (1–5 gamma/mL): anti-CD14, anti-CD16, anti-CD3, anti-CD56, and anti-CD19. The samples were analyzed using the CytoFLEX-S flow cytometer (Beckman Coulter, Brea, CA, USA). Bi-exponential analysis was performed using CytExpert (Beckman Coulter) and FlowJo V10 (Treestar, Ashland, OR, USA) software. Cells were first gated for singlets (FSC-H vs. FSC-A) and monocytes and lymphocytes based on the physical parameter (SSC-A vs. FSC-A). The events within the gate were analyzed for expression markers. In order to accurately identify the positive dataset, appropriate controls such as isotype, fluorescence minus one (FMO), and single and unstained controls were used. The gate was set for each patient at each sampling and was based on IgG staining.

The monocyte and lymphocyte gates were further analyzed for expression of CD14 and CD16. To accurately define the CD14 monocytes in the enrolled subjects, all CD14^+^ cells were included, providing a wider and more reliable gate. CD14^++^CD16^−^ monocyte cells, CD14^+^CD16^+^ monocytes, and CD16^+^CD14^low^ non-monocyte cells were then identified. Absolute cell count was obtained by multiplying the percentage of CD14/CD16 staining of cell subgroups by the absolute number of monocytes plus lymphocytes, determined using the hematological analyzer. For NK cell analysis, the lymphocyte gate was evaluated for expression of CD3 and CD56. NK cells were defined as CD14^−^CD19^−^CD3^−^CD56^+^ cells. CD3 vs. CD56 plot enabled the identification of CD56^+^CD3^−^ NK cells, CD56^−^CD3^+^ T lymphocytes, and CD3^+^CD56^+^ cells. NK cells were further divided into two subsets (CD3^−^CD56^dim^ (CD56^dim^ NK) and CD3^−^CD56^bright^ NK (CD56^bright^ NK)) on the basis of CD56 density. NK cell count was obtained by multiplying the percentage of CD56^+^CD3^−^ NK cells, CD56^−^CD3^+^ T lymphocytes, and CD3^+^CD56^+^ cell subgroups by the absolute lymphocyte count. CD56^dim^ NK and CD56^bright^ NK cell counts were derived by multiplying the percentage of a given cell subset by the total CD56^+^CD3^−^ NK cell number. For B cell quantification, the lymphocyte gate was evaluated for CD19 expression, and the cell count was obtained by multiplying the percentage by the absolute lymphocyte count determined using the hematological analyzer.

### 2.3. Serum Cytokines Quantification

Serum levels of IL-6, TGF-α, and TNF-α were determined using premixed multiplex human magnetic Luminex assays (R&D, #LXSAHM-11) according to the manufacturer’s instructions. Data acquisition was performed using the Bio-Plex MAGPIX Multiplex reader (BioRad Laboratories, Hercules, CA, USA), which uses Luminex fluorescent-bead-based technology (Luminex, Austin, TX, USA). Serum levels of TGFβ1 (EBioscience, San Diego, CA, USA, #BMS249/4) and IL-15 (Thermo Scientific, Waltham, MA, USA, #BMS2106) were evaluated with enzyme-linked immunosorbent assay (ELISA, Ratavartijankatu, Finland). The optical density was measured spectrophotometrically at a wavelength of 450 nm according to the manufacturer’s instructions. All analyses were performed in duplicate. A 4-PL standard curve was created using MyCurveFit Beta Software (https://mycurvefit.com/, accessed on 11 April 2023).

### 2.4. Body Composition Assessment

DEXA (Dual-Energy X-ray Absorptiometry) was performed to assess body composition (Hologic Inc., Marlborough, MA, USA, QDR 4500 W). Diagnosis of sarcopenia, according to the updated European Working Group on Sarcopenia in Older People (EWGSOP2), was defined as based on appendicular skeletal muscle mass (ASM) over height squared (ASM/heigh^2^, kg/m^2^) < 7.0 kg/m^2^ in men and < 5.5 kg/m^2^ in women [16].

### 2.5. Response and Survival Outcome Assessment

Response to treatment was assessed in accordance with the Response Evaluation Criteria in Solid Tumors (RECIST) version 1.1 [17]. Patients were categorized into two main groups based on their best response to ICIs: (1) disease control group (DC), including complete response (CR), partial response (PR), or stable disease (SD); and (2) progression disease group (PD), including patients with disease progression.

Progression-free survival (PFS) was calculated as the time (in months) from the date of initiation of anti-PD-1/anti-PD-L1 therapy to the date of disease progression or death, whichever occurred first, and was censored at the date of the last visit for patients who were still alive without any documented disease progression. Overall survival (OS) was calculated as the time (in months) from the date of initiation of anti-PD-1/anti-PD-L1 therapy to the date of death due to any cause or to the date of last visit for patients with no confirmation of death. Alive patients at the time of analysis were censored at the last follow-up. Disease control rate (DCR) defined the percentage of patients who achieved CR, PR, or SD. Duration of response (DoR) was calculated as the time (in months) from initiation of treatment to PD or death in patients who had CR or PR as their best response.

Patients were considered under corticosteroid treatment even if they interrupted therapy ≤ 2 weeks before T_0_ evaluation.

### 2.6. Statistical Analyses

Outcome measurements were assessed for normality using the Shapiro–Wilk test, and non-parametric tests were used when violations of parametric test assumptions were evident. Values were expressed as median and interquartile range (IQR).

The Mann–Whitney U test was used to determine whether there were differences between covariates in the different cohorts, and the Wilcoxon signed-rank test was used to compare related samples at the two time points. Categorical variables were examined by chi-square or Fisher’s exact tests, as appropriate. Spearman rank correlation was used to measure the degree of association between two variables. To predict the PFS based on NK concentrations, a linear regression analysis was employed. The Kaplan–Meier (KM) survival analysis was conducted to assess the survival curves, and pairwise log-rank comparisons were conducted to determine which group had significantly different survival distributions. A multivariable Cox regression analysis was performed to determine whether variables that were significant at univariate analysis could affect survival outcomes. A receiver operating characteristic curve (ROC) was performed with the aim of finding the baseline CD3^−^CD56^+^ NK levels that were able to predict an accurate value to discriminate between patients with PD and those without. The analysis of the covariance model provided least-squares mean estimates with 95% confidence interval (CI) adjusted for multiple comparisons. The *p*-values were two-sided for all statistical tests, and *p* < 0.05 was considered to be statistically significant. All statistical analyses were performed with SPSS Statistics version 27.0 (IBM SPSS Statistics Inc., Chicago, IL, USA) and Prism (version 9.0, GraphPad Software, LLC, San Diego, CA, USA).

## 3. Results

### 3.1. Characteristics of the Cohort

The characteristics of the cohort have been extensively reported [15] and are summarized in Table 1. Overall, the final cohort was composed of 47 patients, 27 males (57.4%) and 20 females (42.6%). Based on the best response, 3 patients (6.4%) experienced CR, 9 (19.1%) PR, 18 (38.3%) SD, and 17 (36.2%) PD. The performance status revealed 33 (70.2%) ECOG 0 and 14 (29.8%) ECOG 1 patients. Patients with ECOG 1 showed a higher incidence of PD in the chi-square test (9/14 vs. 8/33, *p* = 0.009), even if no differences were found in PSF (*p* = 0.126) and OS (*p* = 0.136) comparing patients with ECOG 0 and ECOG 1.

The PD-L1 expression on lung biopsies was evaluated in 39 patients.

The median PFS was 8 months (95% CI, 2.2–13.8 months), while the median OS was 14 months (95% CI, 8.4–19.5 months). Overall, 35 PFS events (74.5%) were observed, and 38 patients (80.8%) died during the observation period. The ORR was 25.5%, while the DCR was 63.8%. The median DoR in 30 responder patients was 20 months (95% CI, 14–25.9 months). During the study period, 38 patients (80.9%) had to interrupt treatment with ICIs due to PD or toxicity.

Sixteen patients were under corticosteroid treatment (34%) at T_0_, nine with high dosages (≥10 mg of prednisone or equivalent). In detail, 9 patients assumed prednisone (variable dosage between 5 and 25 mg), 3 patients dexamethasone (variable dosage between 1 and 4 mg), 2 patients betamethasone (4 mg), and 1 patient methylprednisolone (8 mg). Patients with high-dosage corticosteroids showed a higher incidence of PD in the chi-square test (6/9 vs. 11/38, *p* = 0.034), even if no differences were found in PSF (*p* = 0.233) and OS (*p* = 0.097).

No difference in the incidence of PD, PFS, or OS duration was found when comparing the population for histology, type of ICI, treatment line, PD-L1 expression, sex, or smoking status.

Thirty patients (63.8%) underwent the T_1_ evaluation, while the others were unable to undergo it due to death, clinical conditions, or hospitalization.

### 3.2. Baseline Evaluation (T_0_)

The data on whole blood count, inflammatory markers, PBMCs and cytokines are reported in Table 2 and Table 3.

A subgroup analysis based on PD or DC was performed. No differences were found in the inflammatory markers and cytokine assessment between the two groups (Table 2).

Among PBMCs, significant differences among the two groups were found in the percentage and absolute count of CD3^−^CD56^+^ NK cells (27.8 cells/µL (4.6; 57.6) vs. 127 cells/µL (58; 210.1), (*p* < 0.001)) (Figure 1 and Figure 2, Table 3).

The Spearman test showed a positive correlation between PFS and OS with CD3^−^CD56^+^ NK cells (*p* < 0.001; *p* = 0.003). Simple linear regression was used to test if CD3^−^CD56^+^ NK cells significantly predicted PFS and OS. A longer OS was predicted by higher levels of CD3^−^CD56^+^ NK cells (R^2^ = 0.25, F (1.41) =13.684, *p* < 0.001) (Figure 3).

A ROC curve analysis, performed to find an accurate CD3^−^CD56^+^ NK value to discriminate between patients with PD and DC (as best response), revealed that a cut-off level of 53.4 cells/µL had a sensitivity of 75% and a specificity of 77.8% (AUC: 0.831, 95% CI: 0.711–0.951, *p* < 0.001) (Figure 4). Based on this cut-off value, Kaplan–Meier curves were performed. The results showed that patients with CD3^−^CD56^+^ NK < 53.4 cells/µL had shorter survival, both PFS and OS (PFS: 3 months, 95% CI: 1.6–4.3 vs. 15 months, 95% CI: 10.7–19.3, *p* < 0.001; OS: 6 months, 95% CI: 3.9–8.1 vs. 20 months, 95% CI: 7.6–20.3, *p* < 0.001) (Figure 5).

To determine whether CD3−CD56^+^ NK can affect survival outcomes, together with other risk factors for worse survival that were significant in the univariate analysis (ECOG and high corticosteroid dosage), a multivariable Cox regression analysis was applied. The results showed that CD3−CD56^+^ NK (HR 0.988; 95% CI 0.981–0.994, *p* < 0.001) were independent prognostic factors for shorter PFS together with high corticosteroid dosage (HR 2.554; 95% CI 1.026–6.361, *p* = 0.044). The same factors were confirmed for shorter OS: CD3^−^CD56^+^ NK (HR 0.992; 95% CI 0.987–0.997, *p* < 0.001) and high corticosteroid dosage (HR 2.42; 95% CI 1.002–5.843, *p* = 0.049) (Table 4).

Among NK cells, both percentage and absolute numbers of CD56^dim^NK and CD56^bright^NK were lower in the PD group: CD56^dim^NK 16.8 cells/µL (2.1; 49) vs. 85.9 cells/µL (20.1; 146.6), *p* = 0.003; CD56^bright^NK 6.7 cells/µL (1.5; 17.6) vs. 18.1 cells/µL (4.2;49), *p* = 0.047. No significant differences were found in the CD56^bright^/CD56^dim^NK cell ratio (Table 3, Figure 1 and Figure 2).

No differences were found in any PBMC subpopulations analyzed when comparing patients who were under corticosteroids with those who were not (even if considering only patients with a dosage ≥ 10 mg of prednisone or equivalent). No differences were found in PBMCs in patients who had received chemotherapy prior to beginning immunotherapy either, except for CD56^dim^NK cells/µL (39.1 (9.1; 78.58) vs. 93.2 cells/µL (30.6;152), *p* = 0.04) and B lymphocytes (59.3 (37.8; 120.1) vs. 121.6 cells/µL (73.9; 258.3), *p* = 0.038), which were significantly lower in the pre-treated patients.

### 3.3. T_1_ Evaluation

At the first radiological evaluation (T_1_), the PD group showed higher neutrophil/lymphocyte ratio (NLR) (*p* = 0.008), leukocyte/lymphocyte ratio (LLR) (*p* = 0.004), CPR (*p* = 0.01), and fibrinogen (*p* = 0.015), as well as lower lymphocyte (*p* < 0.001), transferrin (*p* = 0.015), and TGF-α (*p* = 0.034) levels compared to the DC group (Table 2).

The PBMC subset analysis showed lower CD3^−^CD56^+^ NK (30.4 cells/µL (15.3; 53.8) vs. 138.1 cells/µL (94.5; 223.6), *p* < 0.001) and CD56^dim^NK (29 cells/µL (9;41.1) vs. 100.3 cells/µL (64.1; 145.2), *p* < 0.009) levels in PD patients compared to the DC group. A difference in B lymphocytes, which were fewer in the PD group (*p* = 0.044), was also found (Figure 1 and Figure 2, Table 3).

### 3.4. Longitudinal Evaluation (T_1_ vs. T_0_)

A longitudinal comparison between T_1_ and T_0_ was also performed (Table 2 and Table 3). A reduction in fibrinogen (*p* = 0.017) and ferritin (*p* = 0.036) levels was shown at T_1_ in the overall population. Among PBMCs, only CD56^bright^NK significantly increased at T_1_ (*p* = 0.006).

In the subgroup analysis for PD or DC, ESR increased in the PD group (*p* = 0.042), while fibrinogen decreased in the DC group (*p* = 0.023). Moreover, differences were found in the DC group for both monocytes and NK cells. The former, in fact, decreased at T_1_ as total monocytes (CD3^−^CD14^+^ 331.5 cells/µL (179.2; 536.2) vs. 427.6 cells/µL (280.7; 620.3), *p* = 0.031) and in the subclass of CD3^−^CD14^++^CD16^−^ (331.5 cells/µL (179.2; 536.2) vs. 427.6 cells/µL (280.7; 620.3), *p* = 0.035). Conversely, the total number of NK cells significantly increased at T_1_ (CD3^−^CD56^+^ 138.1 cells/µL (94.5; 223.6) vs. 127 cells/µL (58; 210), *p* = 0.025) together with the subclass of CD56^bright^NK (27.4 cells/µL (10.4; 85.7) vs. 18.1 cells/µL (4.2; 49), *p* = 0.034). Finally, in the PD group, CD56^dim^NK cells increased at T_1_ (29 cells/µL (9;41.1) vs. 16.8 cells/µL (2.1; 49), *p* = 0.043). However, no differences were found between T_0_ and T_1_ in both groups for the CD56^bright^/CD56^dim^ ratio (Table 3, Figure 6).

In regard to the cytokine analysis, the DC group showed a consistent increase in IL-15 concentrations (4.9 pg/mL (3.6; 10) vs. 1.2 pg/mL (0.4; 2.4), *p* < 0.001) (Table 2, Figure 5). TGF-β significantly decreased in the overall population at T_1_ vs. T_0_ (6.1 pg/mL (4.0; 7.6) vs. 4.1 pg/mL (3.5; 5.1), *p* < 0.001) (Table 2).

We previously demonstrated that sarcopenia is associated with shorter PFS and increases the risk of PD. It is also linked with increased inflammatory markers and pro-inflammatory cytokines [15].

Interestingly, patients with sarcopenia showed lower CD3^−^CD56^+^ NK (28.3 cells/µL (95% CI: 4.7–76.1) vs. 114.6 cells/µL (95% CI: 44.3–183.6), *p* = 0.004) and CD56^dim^NK (13.2 cells/µL (95% CI: 2.1–55.2) vs. 89.4 cells/µL (95% CI: 28.3–147.2), *p* < 0.001) values than patients without sarcopenia, while no differences were found for CD56^bright^NK and other PBMC populations. A chi-squared test showed that 12 patients with sarcopenia (63.2%) presented CD3^−^CD56^+^ NK levels below the cut-off of 53.4 cells/µL, compared with 7 non-sarcopenic patients (36.8%) (*p* = 0.03).

## 4. Discussion

Over the past few years, ICIs have dramatically improved survival in many patients with otherwise untreatable cancers. In this study, we aimed to explore (in a basal and longitudinal evaluation) if immune cells could be predictive of better outcomes and long-term survival in advanced NSCLC patients undergoing ICI treatment and to detect eventual connections with sarcopenia, a known risk factor for PD.

Our main findings were the following: (1) CD3^−^CD56^+^ NK basal levels were higher in patients with DC compared to patients with PD; (2) higher NK basal levels predicted a longer OS; conversely, lower NK values represented a risk factor for PD: a value lower than 53.4 cells/µL indicated a shorter survival time; (3) after 3 months of ICI treatment, NK cells significantly increased in the DC group; (4) sarcopenic patients showed lower basal CD3^−^CD56^+^ NK cell values; (5) IL-15 increased in the longitudinal evaluation in the DC group while remaining stable in the PD group.

The positive clinical impact of innate cells has been documented in several tumors following both chemotherapy and ICIs [18,19,20]. In genetically engineered mouse models of lung cancer, it has been observed that therapy-induced cross-talk between macrophages and regulatory T cells sustains tumor resistance to ICIs [10]. Interestingly, NK-depleted mice showed a completely ineffective PD-1/PD-L1 blockade [21]. The importance of NK cells in tumor response depends on the recognition of activating receptors on cancer cells, which are able to rapidly trigger target cell lysis and release pro-inflammatory cytokines, regardless of antigenic presentation [11,12,13,22,23,24].

In our population, patients with higher circulating NK cell levels showed longer survival. Moreover, NK levels were found to be independent predictor variables for both PFS and OS in the Cox regression analysis. These findings confirmed previous results in the literature in the setting of NSCLCs treated with ICIs [14,25,26,27,28,29,30,31,32,33]. PD-1 is expressed on the NK surface, especially on the CD56^dim^NK cell subtypes, therefore enhancing their cytotoxic effect [34], cytokine production, and degranulation [35]. Specifically, we identified 53.4 cells/µL as a cut-off for NK cells to predict better (for higher/equal values) or worse (for lower values) survival. These data can be very useful in evaluating the possible therapeutic benefit of the individual patient. Two previous studies, both with small sample sizes, also identified the NK cut-off [29,33].

CD56^bright^NK cells are considered efficient cytokine producers and are the main NK cell population infiltrating cancer tissues, while the CD56^dim^NK cell subset (the most mature and 90% of peripheral blood NK cells) is able to mediate a strong cytotoxic response upon the engagement of activating receptors and even to exert antibody-dependent cell-mediated cytotoxicity (ADCC) [36]. Among NK cells, we detected that both CD56^dim^NK and CD56^bright^NK were higher in patients with DC compared to patients with PD. Nevertheless, it should be noted that overall NK levels are significantly lower in our population than in non-oncological control patients, as already described in previous studies [37,38,39]. Indeed, the literature already indicates that oncologic patients have defective NK cells [40] and, at the same time, that impaired NK function leads to a higher risk of developing different types of cancer [41]. Our results, in line with many reports, indicated that NK activity is reduced in patients with advanced cancer [42].

Interestingly, no significant differences in the subgroup analysis were observed in full blood count and inflammatory markers. Therefore, routine blood tests do not allow for the identification of patients with a better or worse prognosis at baseline.

Conversely, after 3 months of treatment, the two subgroups showed important differences in inflammatory markers such as NLR, LLR, CRP, and fibrinogen, which were higher in the PD group. Interestingly, these variables have already been associated with worse outcomes in previous studies [43,44,45]. In patients with a better prognosis, lymphocytes increased in number as a response to therapy. Regarding PBMCs, NK cells remained higher in the DC group (specifically CD56^dim^), as previously reported in the literature [28].

The longitudinal evaluation revealed some interesting data. Overall, after ICI administration, ferritin and fibrinogen levels decreased at T_1_, with an even more significant reduction in fibrinogen in the best prognosis group. In line with these results, TGF-β, an immunosuppressive factor, significantly decreased in response to treatment in our cohort as a whole as well as in the two subgroups [46]. TGF-β plays a biphasic role in cancer, acting as a tumor suppressor in the initial stages by suppressing cell proliferation and inducing apoptosis; at the same time it protect cancer cells by suppressing anti-tumor immune responses in the advanced phases [47,48]. The total number of NK cells significantly increased in the DC group in the longitudinal analysis. Therefore, patients with a better response, who already had higher basal NK levels, also showed an increase in the number of these cells at T_1_, while no rise was recorded in patients with PD. Among NK subclasses, even if both levels increased in the DC group, only CD56^bright^NK reached statistical significance, but no differences were found in the CD56^bright^/CD56^dim^ ratio. Considering that CD56^bright^NK cells represent the main NK cell population in cancer tissue, their increase could reflect a better response to treatment. However, despite this result, NK levels remained significantly below the normal levels found in non-neoplastic populations [37,38,39]. In the PD group, instead, CD56^dim^NK cells increased compared to T_0_, even if they reached values below half of the baseline values observed in the DC group.

The total monocyte value, in particular the classical CD14^++^CD16^−^, decreased in DC at T_1_. Classical CD14^++^CD16^−^ macrophages are generally identified as inflammatory monocytes, able to release different proinflammatory cytokines, while non-classical CD14^+^CD16^++^ play an anti-inflammatory role [49,50]. Monocytes’ role in cancer settings is twofold: on the one hand, classical monocytes have pro-tumoral functions (metastatic cell seeding, suppression of T cell function, recruitment of regulatory T cells, angiogenesis, and extracellular matrix remodeling). At the same time, they could also have a protective function due to their cytotoxicity and antigen presentation abilities [27,51]. In fact, their concentrations have been associated with better survival outcomes in patients with advanced NSCLC treated with pembrolizumab [32,50]. In our cohort, we hypothesized that the reduction in classical monocytes could reflect the reduction in the inflammatory state in these patients [52,53].

The alteration of the immune system is likely to play a rather important role in the progression of sarcopenia. As we have already demonstrated, sarcopenia is a predictive factor of worse survival outcomes in oncologic patients, and it can predict a worse response to ICI treatment with an eight-fold higher risk of progression disease than non-sarcopenic patients [15]. In this analysis, we found that sarcopenic patients had lower NK levels compared to non-sarcopenic patients. These results have been confirmed by several studies and recent meta-analyses [54,55]. However, the link between NK cells and sarcopenia remains relatively unexplored. While there are some data on NK cell changes in aging (with a significant reduction in spontaneous cytotoxic capacity being numerically decreased, normal, or even increased) [56,57,58], no specific data exist in the context of sarcopenia, especially in cancer patients. Interestingly, in our cohort, patients with sarcopenia had, in most cases, NK values below the cut-off that predicted PD.

In order to explore the link between sarcopenia, NK function, and survival clinical outcomes, we also analyzed the levels of IL-15, a myokine that is largely produced by normal skeletal muscle tissue [59,60] and is required for the development, maturation, and survival of NK cells, together with other cytokines [61,62]. Evidence has demonstrated that IL-15 deficiency is typical of the sarcopenic state and, at the same time, that IL-15 is essential for better activation of NK cells.

Understanding the complete relationship between NK function, IL-15, sarcopenia, and PD is a very intriguing and complicated challenge. We assumed that the reduction in lean mass due to sarcopenia leads to lower levels of IL-15 and therefore NK cells. However, from our results, probably because of the small sample size, we cannot fully confirm this hypothesis. We proved that basal NK levels were reduced in patients with sarcopenia, but IL-15 levels were not different. Certainly, an impact of IL-15 on the number of NK cells was proven from our longitudinal results: the levels of IL-15 increased in patients with DC, suggesting a possible role in the immune response to tumors during ICI treatment, probably enhancing NK functions [60]. At the same time, we stated that sarcopenic patients had worse survival outcomes. Moreover, from the Cox regression analysis, another variable that significantly influences survival outcomes is high-dosage corticosteroids, which certainly have an impact on PBMCs and cytokine production.

These considerations are complicated by the fact that sarcopenia, tumor progression, and immune and cytokine functions are influenced by multiple other factors, some of which are not yet fully understood. However, if confirmed by larger studies, the connection between NK function, IL-15, sarcopenia, and tumor progression may suggest some interesting perspectives.

In particular, recent preclinical [63,64] and clinical studies [65,66,67] have laid the foundations for the development of new therapeutic scenarios, especially for patients who do not respond to immunotherapy but who could benefit from IL-15 superagonist administration to improve the action of ICIs themselves. Moreover, a specific treatment (or prevention) of sarcopenia can contribute to improving survival outcomes in these patients.

The main points in favor of this study are its prospective nature and the long duration of follow-up, which give a more precise evaluation in terms of survival, allowing us to draw conclusions despite the small sample size. Moreover, even if circulating NK cells do not necessarily reflect their action and concentration in the tumor site [68], their identification with a simple blood test could represent a very useful biomarker to use in daily clinical practice to hypothesize patient response to treatment.

The present study certainly has some important limitations. First, the sample size is very small, limiting the interpretation of results; therefore, studies with a larger cohort are definitely needed. Moreover, only 30 patients out of the whole cohort reached the second time point. Second, even if the population is homogeneous for the type, stage of cancer (all patients with stage IV NSCLC), and PS, the data are not homogeneous for the type of treatment and line of therapy, including patients who have used both PD-1/PD-L1 inhibitors as first-, second-, and third-line therapy.

## 5. Conclusions

In this proof-of-concept, prospective longitudinal observational study, we demonstrated that NSCLC patients treated with anti-PD-1/PD-L1 showed worse survival outcomes if they had a low baseline level of circulating NK cells. Low NK values during ICI treatment (3 months after the first ICI administration) are associated with a worse response to treatment. The longitudinal data revealed that patients with DC had a significant increase in NK cells, while NK concentrations did not change in patients with PD. At the same time, the levels of monocytes appeared to be reduced, probably due to a reduction in the inflammatory state.

We acknowledge that these conclusions were derived from a limited sample of patients. Studies with larger sample sizes are definitely needed to confirm these results.

However, taken all together, these findings can help identify the best candidates for immunotherapy with a simple blood draw. Moreover, they can suggest new therapeutic strategies for non-responder patients.

## Figures and Tables

**Figure 1 cancers-15-03592-f001:**
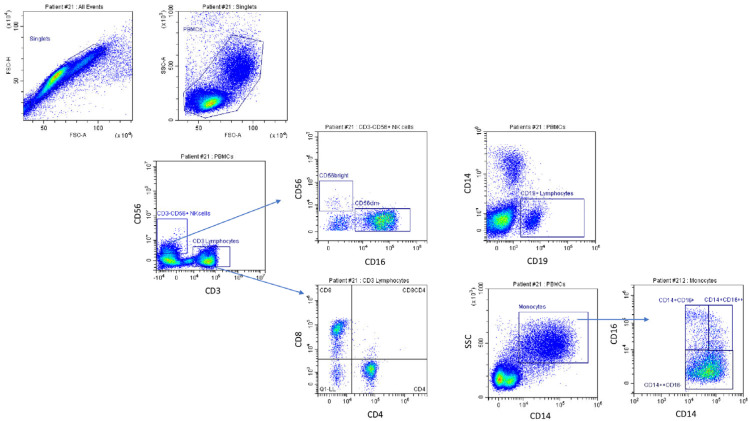
Gating strategy for natural killer cells, lymphocytes, and monocytes. Cells were first gated for singlets (FSC-H vs. FSC-A) and monocytes and lymphocytes based on the physical parameter (SSC-A vs. FSC-A) for peripheral blood mononuclear cells (PBMCs). The gated cell population was analyzed for expression of CD3 and CD56 surface markers to identify natural killers (CD56^+^CD3^−^). CD3^−^CD56^bright^ and CD3-CD56^dim^ cell subsets were identified by gated cells on the basis of CD56 density and CD16 density. The CD8 and CD4 lymphocytes were identified by gate cells on the CD3^+^CD56^−^ T lymphocyte gate. CD19^+^ cells represent B lymphocytes. CD16 and CD14 surface markers were analyzed to identify the subsets of classical monocytes (CD14^+^CD16^−^), non-classical monocytes (CD14^+^CD16^+^), and CD16^+^CD14^−^cells.

**Figure 2 cancers-15-03592-f002:**
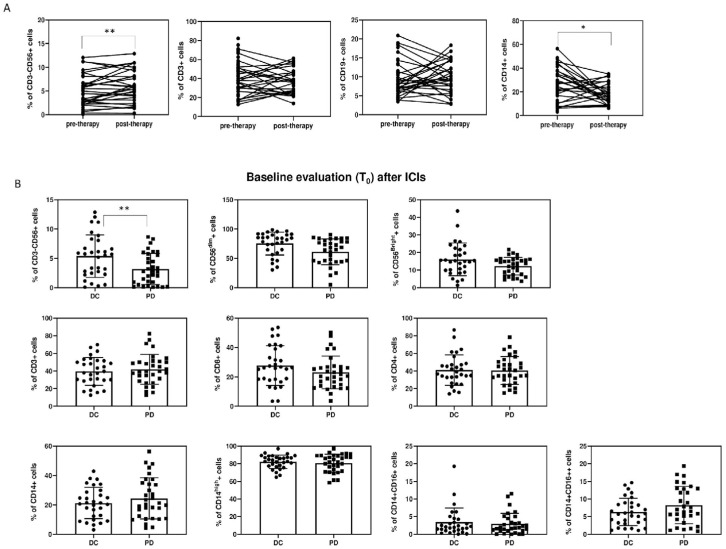
(**A**): Percentages of cells (CD3^−^CD56^+^ NK, CD3^+^CD56^−^ T lymphocytes, CD19^+^ B lymphocytes, and CD14^+^ monocytes) before and after ICI therapy. (**B**): Histogram represents % of cells at baseline evaluation (T_0_). Results are expressed as mean ± SD, * *p* < 0.05, ** *p* < 0.01.

**Figure 3 cancers-15-03592-f003:**
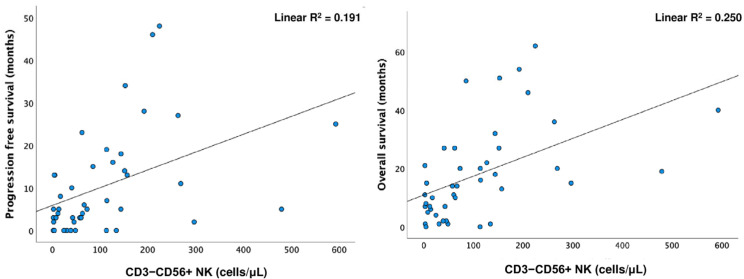
Linear regression for CD3^−^CD56^+^ NK cells and progression-free survival (PFS) (linear R^2^ = 0.191) and overall survival (OS) (linear R^2^ = 0.250).

**Figure 4 cancers-15-03592-f004:**
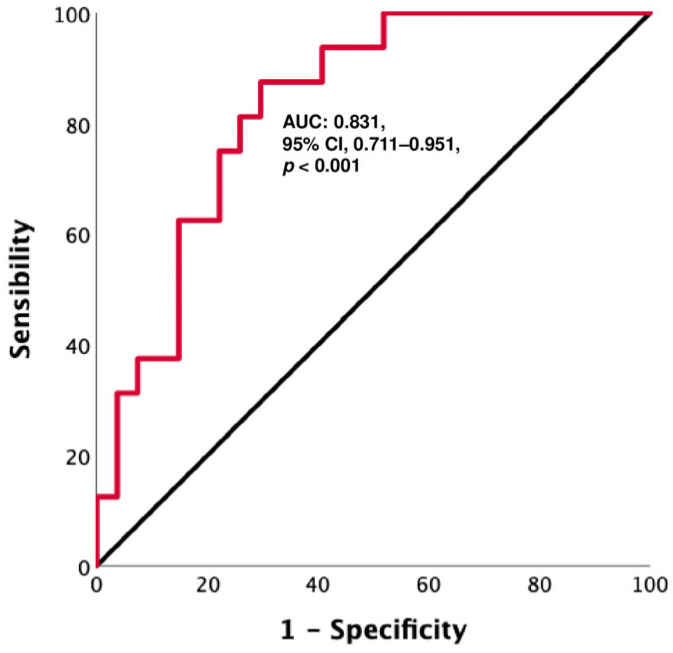
A: ROC curve for CD3^−^CD56^+^ NK cells: a cut-off level of 53.4 cells per µL shows a sensitivity of 75% and a specificity of 77.8% in discriminating between patients with PD and those without (AUC: 0.831, 95% CI 0.711–0.951, *p* < 0.001).

**Figure 5 cancers-15-03592-f005:**
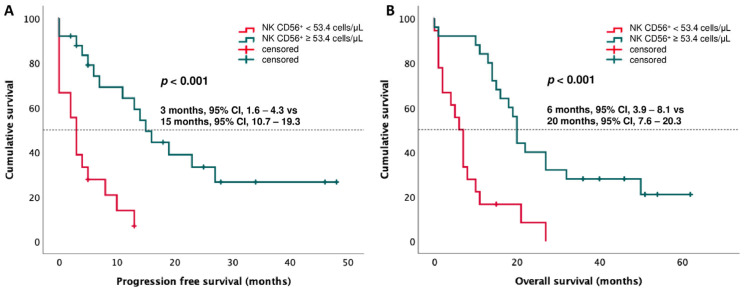
Kaplan–Meier curves based on CD56^+^ NK cell cut-off value of 53.4 cells/µL for (**A**) progression-free survival (PFS) and (**B**) overall survival (OS).

**Figure 6 cancers-15-03592-f006:**
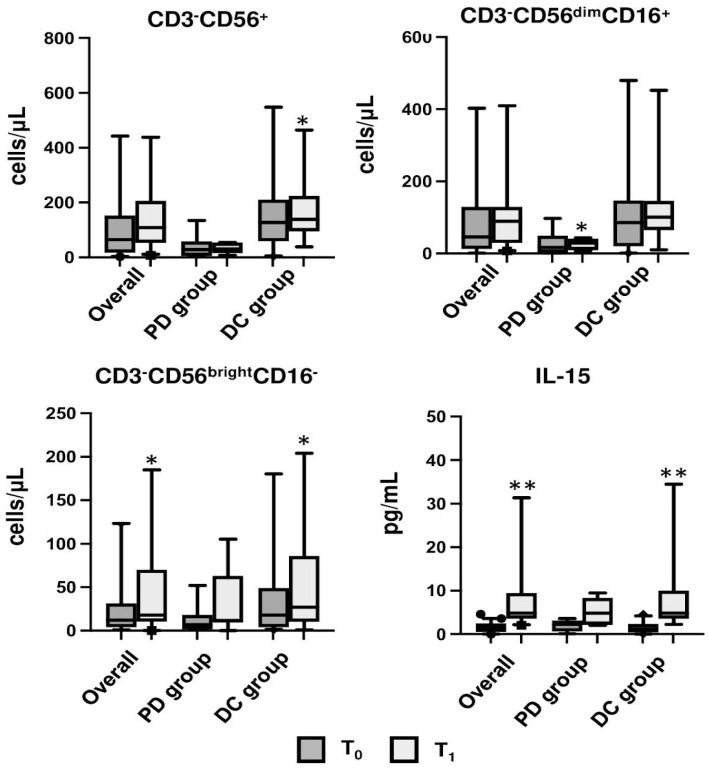
Bar graph showing the median of CD3^−^CD56^+^ NK, CD56^bright^NK, CD56^dim^NK, and IL-15 of the overall population and in the progression disease (PD) and disease control (DC) subgroups compared at T_0_ and T_1_; * indicates *p* < 0.05; ** indicates *p* < 0.001.

**Table 1 cancers-15-03592-t001:** Baseline characteristics of the study population.

Overall, N	47
Age, years, median (IQR)	67 (61;74)
Sex ■ Male, N (%) ■ Female, N (%)	27 (57.4) 20 (42.6)
BMI, Kg/m^2^, median (IQR)	23.9 (20.7; 27.8)
Smoking status ■ Smokers, N (%) ■ Former smokers, N (%) ■ Non-smokers, N (%)	16 (34) 29 (61.7) 2 (4.2)
PS ■ ECOG 0, N (%) ■ ECOG 1, N (%)	33 (70.2) 14 (29.8)
Histotype NSCLC ■ Adenocarcinoma, N (%) ■ Squamous cell carcinoma, N (%) ■ Poorly differentiated carcinoma, N (%) ■ Large cell carcinoma/mixed, N (%)	30 (63.8) 9 (19.1) 5 (10.6) 3 (6.4)
Line of treatment	
■ First line, N (%) ■ Second line, N (%) ■ Third line, N (%)	18 (38.3) 21 (44.7) 8 (17)
Type of ICI ■ Nivolumab, N (%) ■ Pembrolizumab, N (%) ■ Atezolizumab, N (%)	22 (46.8) 18 (38.3) 7 (14.9)
PD-L1 expression ■ <1%, N (%) ■ >1% and < 50%, N (%) ■ ≥50%, N (%) ■ n.a.	10 (21.3) 10 (21.3) 19 (40.4) 8 (17)
Best response ■ CR, N (%) ■ PR, N (%) ■ SD, N (%) ■ PD, N (%)	3 (6.4) 9 (19.1) 18 (38.3) 17 (36.2)
Survival data	
■ Dead, N (%) ■ ORR, N (%) ■ DCR, N (%) ■ PFS, months (95% CI) ■ OS, months (95% CI) ■ DoR, N (%)	38 (80.8) 12 (25) 30 (63.8) 8 (2.2–13.8) 14 (8.4–19.5) 20 (14–25.9)
Sarcopenia, N (%)	19 (40.4)
Corticosteroid, N (%)	13 (27.6)

Continuous variables are expressed as the median (95% CI). Abbreviations: BMI: body mass index; PS: performance status; ECOG: Eastern Cooperative Oncology Group; n.a.: not analyzed; CR: complete response; PR: partial response; SD: stable disease; PD: progression disease; PFS: progression-free survival; OS: overall survival; ORR: overall response rate; DCR: disease control rate; DoR: duration of response.

**Table 2 cancers-15-03592-t002:** Characteristics of white blood cells, inflammatory markers, and cytokines at T_0_ and T_1_.

	T_0_				T_1_			
	Overall (*n* = 47)	PD (*n* = 17)	DC (*n* = 30)	*p*	Overall (*n* = 30)	PD (*n* = 5)	DC (*n* = 25)	*p*
**White blood cells ×10^3^/µL**	8.0 (6.6; 9.2)	8.6 (7.2; 9.2)	7.8 (6.2; 10.1)	0.569	7.4 (6.5; 8.6)	6.6 (5.3; 7.9)	7.5 (6.6; 8.9)	0.284
**Neutrophils ×10^3^/µL**	5.5 (4.4; 6.7)	6.3 (5.2; 6.7)	5.0 (3.9; 7.1)	0.198	4.9 (4.0; 5.5)	4.9 (3.7; 6.0)	4.8 (4.2; 5.4)	0.928
**Lymphocytes ×10^3^/µL**	1.6 (1.1; 2.0)	1.5 (0.9; 2.1)	1.6 (1.3; 1.9)	0.328	1.6 (1.3; 2.0)	1.1 (0.9; 1.3)	1.7 (1.6; 2.1)	<0.001
**Monocytes ×10^3^/µL**	0.5 (0.4; 0.7)	0.5 (0.3; 0.6)	0.6 (0.4; 0.7)	0.306	0.5 (0.4; 0.6)	0.4 (0.3; 0.5)	0.5 (0.4; 0.7)	0.099
**Eosinophils ×10^3^/µL**	0.11 (0.08; 0.27)	0.08 (0.04; 0.13)	0.17 (0.1; 0.31)	0.005	0.14 (0.09; 0.3)	0.15 (0.05; 0.24)	0.14 (0.09; 0.3)	0.694
**Basophils ×10^3^/µL**	0.03 (0.02; 0.04)	0.03 (0.02; 0.04)	0.03 (0.02; 0.04)	0.543	0.03 (0.02; 0.04)	0.02 (0.01; 0.04)	0.03 (0.02; 0.04)	0.129
**NLR**	3.5 (2.5; 5.2)	4.4 (2.6; 6.6)	2.9 (2.1; 4.5)	0.111	2.7 (2.1; 4.2)	4.2 (3.6; 5.7)	2.7 (2.1; 3.2)	0.008
**LLR**	4.8 (3.8; 6.8)	6.1 (4; 7.8)	4.6 (3.7; 5.9)	0.213	4.5 (3.6; 5.6)	5.7 (5.2; 7.5)	4.2 (3.6; 4.9)	0.004
**Inflammatory and iron markers**								
**ESR, mm/h**	56 (24; 86)	59 (35; 107.5)	43 (19.7; 77.5)	0.132	32 (16.5; 70)	75 (40.5; 94.5)	27 (12.2; 60.2)	0.06
**CRP, mg/L**	14.1 (3.6; 42.6)	19.8 (8.9; 55.1)	10.2 (3.1; 31)	0.088	5.30 (17; 29.4)	30.8 (19; 55.3)	3.2 (1.3; 10.5)	0.01
**Fibrinogen, g/L**	4.9 (4.0; 6.1)	5.9 (4.3; 6.3)	4.7 (3.8; 5.8)	0.112	3.9 (3.3; 5.6) *	5.8 (4.7; 6.8)	3.7 (3.3; 4.9) *	0.015
**Ferritin, µg/L**	243 (166; 394)	262 (189.5; 409.7)	240 (146.2; 388.7)	0.782	184.5 (77.2; 364.5) *	254 (59.5; 346.5)	182 (86.5; 367)	0.705
**Transferrin, g/L**	2.3 (2.0; 2.6)	2.2 (2.0; 2.5)	2.3 (2.0; 2.6)	0.784	2.3 (2; 2.5)	1.9 (1.5; 2.2)	2.4 (2.2; 2.6)	0.015
**Cytokines**								
**IL-6, pg/mL**	5.8 (3.2; 17.0)	12.7 (3.0; 20)	5.8 (3.5; 12.9)	0.344	5.4 (2.3; 7.9)	5.0 (2.8; 14.8)	5.6 (2.3; 7.6)	0.933
**TNF-α, pg/mL**	4.3 (2.6; 6.0)	5.2 (3.5; 6.0)	3.5 (2.1; 5.6)	0.276	4.3 (2.6; 6.4)	4.3 (2.1; 5.2)	4.3 (2.8; 6.8)	0.414
**TGF-β, pg/mL**	6.1 (4.0; 7.6)	5.3 (3.8; 8.1)	6.3 (4.2; 7.6)	0.654	4.1 (3.5; 5.1) **	3.7 (3.2; 6) *	4.1 (3.5; 5) *	0.880
**TGF-α, pg/mL**	6.1 (3.6; 15.3)	6.9 (4.0; 16.8)	5.3 (3.2; 13.9)	0.511	6.1 (3.6; 13.0)	2.7 (1.7; 5.7)	9.2 (4.0; 15)	0.034
**IL-15, pg/mL**	1.4 (0.6; 2.5)	2.3 (0.6; 3.1)	1.2 (0.4; 2.4)	0.253	4.9 (3.6; 9.5) **	4.9 (2.2; 8.4)	4.9 (3.6;10) **	0.358

At each time point, data are reported for the overall population and for subgroups based on best response: progression disease (PD) and disease control (DC). Data are expressed as median (IQR). Comparisons between the two groups were performed at each time point using Mann–Whitney U test. Wilcoxon test was used for longitudinal evaluation (T_1_ vs. T_0_): * indicates *p* < 0.05; ** indicates *p* < 0.001 for longitudinal evaluation.

**Table 3 cancers-15-03592-t003:** Characteristics of PBMC at T_0_ and T_1_.

	T_0_				T_1_			
	Overall (*n* = 47)	PD (*n* = 17)	DC (*n* = 30)	*p*	Overall (*n* = 30)	PD (*n* = 5)	DC (*n* = 25)	*p*
**Total Monocytes** **CD14^+^, cells/μL**	427.6 (280.7; 620.3)	354.6 (247.1; 585.5)	436.2 (291.6; 716.0)	0.501	331.5 (179.2; 536.2)	392.2 (211.2; 573.8)	325.8 (176.8; 567.6) *	0.629
CD14^++^ CD16^−^, cells/μL	308.8 (168.9; 510.0)	299.9 (113.3; 438.0)	314.0 (168.9; 629.4)	0.453	268.4 (144.4; 396.6)	326.3 (170.7; 483.0)	253.5 (141.1; 407.9) *	0.581
CD14^+^ CD16^+^, cells/μL	7.0 (2.9; 22.4)	6.4 (3.9;16.9)	7.8 (1.9;25.3)	0.959	5.5 (3.1;15.3)	11.5 (3.1;16.2)	5.2 (3.1;15.8)	0.783
CD14^+^ CD16^++^, cells/μL	20.8 (14.6; 53.2)	18.9 (16.0; 67.3)	23.4 (13.9; 53.2)	1.000	25.9 (19.5; 43.8)	27.5 (10.7; 31.0)	24.4 (19.5; 45.8)	0.446
**T lymphocytes** **CD3^+^ CD56^−^, cells/μL**	822.4 (526.8; 1194.4)	773.4 (455.7; 1147.8)	911.6 (587.3; 1345.1)	0.437	741.6 (509.6; 833.8)	370.4 (234.3; 926.1)	744.6 (636.0; 841.8)	0.265
CD4^+^, cells/μL	284.8 (151; 551)	215.6 (138.1; 612.9)	320.6 (156.9; 543.7)	0.445	264.8 (160.4; 307.8)	167.6 (61.0; 289.1)	265.3 (177.3; 312.3)	0.275
CD8^+^, cells/μL	234.6 (105.2; 316.9)	153.5 (96.4; 298.3)	273.9 (132.9; 334.5)	0.133	179.4 (91.6; 288.7)	125.5 (54.5; 251.5)	184.0 (98.6; 323.9)	0.406
**NK cells** **CD3^−^CD56^+^, cells/μL**	63.6 (17.5; 151.7)	27.8 (4.6; 57.6)	127.0 (58; 210)	<0.001	108.6 (53.6; 205.9)	30.4 (15.3; 53.8)	138.1 (94.5; 223.6) *	< 0.001
CD56^dim^, cells/μL	45.5 (13; 129)	16.8 (2.1; 49.0)	85.9 (20.1; 146.7)	0.003	89.2 (29.4; 129.0)	29.0 (9.0; 41.1) *	100.3 (64.2;145.2)	0.009
CD56^bright^, cells/μL	12.0 (3.6; 31.2)	6.7 (1.5; 17.6)	18.1 (4.2; 49.0)	0.047	17.9 (10.5; 69.8) *	12.0 (10.0; 48.5)	27.4 (10.4; 85.7) *	0.357
Ratio CD56^bright^/CD56^dim^	0.20 (0.09; 1.13)	0.49 (0.11; 1.27)	0.16 (0.06; 0.77)	0.239	0.24 (0.07;0.89)	0.99 (0.78;1.12)	0.17 (0.06; 0.77)	0.462
**B lymphocytes** **CD19^+^, cells/μL**	92.6 (39.5;135.4)	106.0 (44.4;203.6)	61.6 (39.0;125.8)	0.468	69.2 (40.9;93.4)	41.4 (7.0;65.5)	78.9 (42.1;96.4)	0.044

At each time point, data are reported for the overall population and for subgroups based on best response: progression disease (PD) and disease control (DC). Data are expressed as median (IQR). Comparisons between the two groups were performed at each time point using Mann–Whitney U test. Wilcoxon test was used for longitudinal evaluation (T_1_ vs. T_0_): * indicates *p* < 0.05 for longitudinal evaluation.

**Table 4 cancers-15-03592-t004:** Cox regression analysis predicting independent risk factors for shorter progression-free survival (PFS) and overall survival (OS) of significant variables at univariate analysis (ECOG, high-dosage corticosteroid treatment, and NK CD3^−^CD56^+^ levels).

		B	SE	Wald	df	*p*	OR	95% CI for OR	
								Lower	Upper
	ECOG	0.508	0.399	1.623	1	0.203	1.662	0.761	3.634
PFS	High-dosage corticosteroid	0.938	0.465	4.060	1	0.044	2.554	1.026	6.361
	NK CD3^−^CD56^+^ (cells per μL)	−0.012	0.003	13.673	1	<0.001	0.988	0.981	0.994
	ECOG	0.465	0.358	1.685	1	0.194	1.592	0.789	3.215
OS	High-dosage corticosteroid	0.884	0.450	3.865	1	0.049	2.420	1.002	5.843
	NK CD3^−^CD56^+^ (cells per μL)	−0.008	0.003	11.082	1	<0.001	0.992	0.987	0.997

## Data Availability

The data presented in this study are available on request from the corresponding author. The data are not publicly available due to privacy-related issues.

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
