# Peer review of "Circulating Natural Killer Cells as Prognostic Value for Non-Small-Cell Lung Cancer Patients Treated with Immune Checkpoint Inhibitors: Correlation with Sarcopenia"

_cancers, 2023, doi:10.3390/cancers15143592_

Round 1
Reviewer 1 Report
I read with great interest the paper entitled: “Circulating NK cells as prognostic value for NSCLC patients 2 treated with immune checkpoint inhibitors: correlation with 3 sarcopenia”. This is a very well written manuscript. The introduction is concise and adequate. I would have some comments and suggestions:
Regarding Methodology
1. Since the patients received three different ICIs (nivo, pembro and atezo) and different lines, it is important to describe in Methods the PD-L1 staining performed on lung tissue by a pathologist with expertise in lung pathology. Current clinical trials indicate that PD-L1 expression is not necessary for Nivo treatment. In contrast, Pembro in first line require PD-L1 expression > 50%. Since the cohort included patients in stage IV, probably the samples were small biopsies. Specify which samples were considered suitable for PD-L1 staining. How many malignant cells were accessible. How was the anti-PD1 antibody and correspondent platform. How was PD-L1 expression assessed?
2. According to PD-L1 expression on biopsy specimens, I suggest to separating patients in three groups: negative (less than 1%), intermediate (between 1 and 50%) and high (more than 50%).
3. Comment the reason that CD3/NK/PD-1 were not evlauated in the study
4. It is amazing the ROC curves results in a very small cohort of patients (75% and a specificity of 77.8% in discriminating between PD or not (AUC: 0.831, 95% CI 0.711–14 0.951) p<0.001). Please, comment.
5. Why the authors didn´t construct ROC curves with state variable represented by response to Nivo, Pembro e Atezo?
6. I suggest stratifying the histologic types as squamous cell carcinomas and non-small cell carcinomas to verify the impact of NK cells.
7. A Cox regression model should be performed to assess the impact of the different variables on OS. The variables selected for inclusion in the model should be those that were significant in the univariate survival analysis: ECOG, stage, indication for treatment (chemotherapy, radiotherapy, corticosteroid), type of immunotherapy (Nivo, Pembro and Atezo, best response, and CD3−CD56 +NK.

The English language is reasonable.
Author Response
We would like to thank the Editor, the Associated Editor and the Referees for appreciating our paper and for having raised some interesting points that have allowed us to improve the revised MS.
We believe that now the clarity of the manuscript has been significantly improved. We have been able to address all the comments raised and the detailed point-by-point rebuttal is listed below.
We also updated Tables and figures so as to eliminate supplementary material.
Reviewers' Comments to Author:
Referee: 1
I read with great interest the paper entitled: “Circulating NK cells as prognostic value for NSCLC patients 2 treated with immune checkpoint inhibitors: correlation with 3 sarcopenia”. This is a very well written manuscript. The introduction is concise and adequate. I would have some comments and suggestions:
Regarding Methodology
- Since the patients received three different ICIs (nivo, pembro and atezo) and different lines, it is important to describe in Methods the PD-L1 staining performed on lung tissue by a pathologist with expertise in lung pathology. Current clinical trials indicate that PD-L1 expression is not necessary for Nivo treatment. In contrast, Pembro in first line require PD-L1 expression > 50%. Since the cohort included patients in stage IV, probably the samples were small biopsies. Specify which samples were considered suitable for PD-L1 staining. How many malignant cells were accessible. How was the anti-PD1 antibody and correspondent platform. How was PD-L1 expression assessed?
R: Thanks for these interesting observations. PD-L1 analysis was performed in 39/47, because, at time of enrollment, this analysis was not requested to start second line ICI treatment. We added this information in the text.
The samples were small biopsies on lung tissue and were analysed by a pathologist of our Hospital with expertise in lung and PD-L1. Serial sections were obtained from each paraffin block for immunohistochemical evaluation of PD-L1 expression neoplastic cells and. PD-L1 immunostains were performed with one of the antibody clones approved for diagnostic assay (PD-L1 SP263, rabbit monoclonal antibody, Ventana Medical Systems Inc, Tucson, USA). Immunohistochemical staining was performed with BenchMark GX immunoautomate (Ventana Medical Systems Inc, Tucson, USA), OptiView DAB IHC Detection Kit and OptiView Amplification Kit (Ventana Medical Systems Inc, Tucson, USA). Relevant positive controls (human tonsils and placenta) were used for each run of staining. The expression of PD-L1 was evaluated on all tumour cells. A minimum of 200 neoplastic cells were present in each biopsy sample. A positive stain was defined as the presence of membrane staining, either strong or weak, complete or incomplete, in a percentage of cells ≥ 1%, that is the threshold reported for clinical response to PD-L1 inhibitors in non small-cell lung cancer.
We summarised this methodology in the material and method section.
- According to PD-L1 expression on biopsy specimens, I suggest to separating patients in three groups: negative (less than 1%), intermediate (between 1 and 50%) and high (more than 50%).
R: Thank you for your suggestion. We have identified the number of patients belonging to the individual groups according to the subdivision you recommended and we added the data in the text (page 5,6) and in table 1. No differences were found in PFS and OS when stratifying the population for the 3 groups of PD-L1 expression (See Results, 3.1 Characteristic of the Cohort).
- Comment the reason that CD3/NK/PD-1 were not evaluated in the study
R: Thank you for the comment. We have analysed the portion of PD-1 expressing cells in CD3+ and CD3-CD56+ cells at basale level. Consistent with previous observation (S. Ottonello et al. Front. Immunol., 2020; Trefny MP et al. Cancer Immunology 2020; Niu C. et al Int J Med Sci 2020; Gascon-Ruiz et al Cancers 2023 and others) on the negative impact of PD-1 expression on patients' OS, we observed in a small number of patients that PD patients differed from DC patients by displaying higher level of PD-1 on almost all lymphocyte subpopulations examined (CD3+, CD8+, CD4+, CD3-CD56+). Unfortunately the small sample size can’t have scientific merit to add our observation result in the manuscript.
- It is amazing the ROC curves results in a very small cohort of patients (75% and a specificity of 77.8% in discriminating between PD or not (AUC: 0.831, 95% CI 0.711–14 0.951) p<0.001). Please, comment.
R: Thank you for the observation. The data can be explained by the fact that the sample, albeit small, is quite homogeneous in terms of disease stage and performance status, thus reducing the variability of the values.
- Why the authors didn´t construct ROC curves with state variable represented by response to Nivo, Pembro e Atezo?
R: Thank you for this observation. We didn’t construct ROC curves with different treatment because given the low sample size we did not want to further fragment the sample analysis with many subgroup analyses. In any case, we evaluated whether there were significant differences in PBMCs between the various treatments and we did not find any differences, therefore we preferred to proceed with a more homogeneous analysis. Moreover, no differences in PFS and OS were found comparing the different treatments. We had specified these results in the previous paper (PMID: 34944975) but we have now briefly reported them also in this one for greater clarity (See Results, 3.1 Characteristic of the Cohort)
- I suggest stratifying the histologic types as squamous cell carcinomas and non-small cell carcinomas to verify the impact of NK cells.
R: we performed a subgroup analysis by histology but no significant data emerged, probably because most patients had adenocarcinoma and only a minority squamous cell carcinoma.
However, we have specified this data in the text (See Results, 3.1 Characteristic of the Cohort)
- A Cox regression model should be performed to assess the impact of the different variables on OS. The variables selected for inclusion in the model should be those that were significant in the univariate survival analysis: ECOG, stage, indication for treatment (chemotherapy, radiotherapy, corticosteroid), type of immunotherapy (Nivo, Pembro and Atezo, best response, and CD3−CD56 +NK.
R: Thankyou for this very useful suggestion. We performed a COX regression analysis both for PFS and OS including as covariates those who were significative in the univariate analysis: ECOG, high dosage corticosteroids and NK CD56+. NK CD56+ maintains significance in the model along with high-dose corticosteroids. We added these results to the text (see Materials and Methods, Results 3.2 Baseline evaluation, Discussion) and Table 4.
Reviewer 2 Report
Using standard tools such as flow cytometry and measurement of inflammatory markers, the study aims at identifying markers for effect of check point inhibitors (ICI). Furthermore the study included DEXA scans as sarcopenia (a known risk factor for poor response) was also included in the analysis. Baseline and longitudinal (2 samples 3 mths apart) samples were included. . ICI’s are frequently hailed as game changers for NSCLC care but in reality many patients does not seem to derive benefit from ICI so study is highly relevant.
The overall findings were that high NK cells (CD3-CH56+) at baseline leads to better DCR and in patients with response, an increase in NK seen. Conversely, in patients with poor outcome an increase in inflammatory markers was noted.
Material included only 47 patients so chance variation is an issue and conclusions should reflect this.
For corticosteroids, there probably is a dose dependent relation ship with negative impact in ICI so a split in the table between 10 mgs or less vs more than 10 mgs of prednisolone (or equivalent) would be helpful.
ORR was only 25% so to better understand this, it would be helpful if baseline table could include PD-L1% status.
My main reservation with this manuscript is that it is not clear if the observed relation ships are causal or not. I realize that this is a particular challenge taking the complex interactions between immune cells, inflammation and sarcopenia in to consideration. I think the discussion could benefit from some considerations from the authors on possible biological reasons the associations are more or less likely to be real.
Concerning the longitudinal data, it is not really relevant to identify poor responder's after many months of treatment, it is really the data available at treatment start (when you make an actual choice) that is relevant.
Author Response
We would like to thank the Editor, the Associated Editor and the Referees for appreciating our paper and for having raised some interesting points that have allowed us to improve the revised MS.
We believe that now the clarity of the manuscript has been significantly improved. We have been able to address all the comments raised and the detailed point-by-point rebuttal is listed below.
We also updated Tables and figures so as to eliminate supplementary material.
Reviewers' Comments to Author:
Referee: 2
Using standard tools such as flow cytometry and measurement of inflammatory markers, the study aims at identifying markers for effect of checkpoint inhibitors (ICI). Furthermore the study included DEXA scans as sarcopenia (a known risk factor for poor response) was also included in the analysis. Baseline and longitudinal (2 samples 3 months apart) samples were included. . ICI’s are frequently hailed as game changers for NSCLC care but in reality many patients does not seem to derive benefit from ICI so study is highly relevant.
The overall findings were that high NK cells (CD3-CH56+) at baseline leads to better DCR and in patients with response, an increase in NK seen. Conversely, in patients with poor outcome an increase in inflammatory markers was noted.
Material included only 47 patients so chance variation is an issue and conclusions should reflect this.
R: We agree and thank the Reviewer. We know that the sample size is low. We have specified this within the limits of the study but we have strengthened this observation as you suggested (see Conclusions)
For corticosteroids, there probably is a dose dependent relationship with negative impact in ICI so a split in the table between 10 mgs or less vs more than 10 mgs of prednisolone (or equivalent) would be helpful.
R: We thank the reviewer for this very useful suggestion. We carried out an analysis considering only high-dose corticosteroids (> 10 mg prednisone or equivalent): patients with high dosage had higher incidence of PD (p=0.034) (chi-square test) even if no differences were found in PFS and OS at Mann-Whitney. We have modified the text accordingly incorporating these results (See Results, 3.1 Characteristic of the Cohort). Moreover, as suggested by Reviewer 1, we performed a Cox regression analysis and included high dosage corticosteroids among the covariates (see Materials and Methods, Results 3.2 Baseline evaluation, Discussion and Table 4) .
ORR was only 25% so to better understand this, it would be helpful if baseline table could include PD-L1% status.
R: Thank you for this suggestion. We included PD-L1% in Table 1 and in the text accordingly (See Results, 3.1 Characteristic of the Cohort).
My main reservation with this manuscript is that it is not clear if the observed relationships are causal or not. I realize that this is a particular challenge taking the complex interactions between immune cells, inflammation and sarcopenia into consideration. I think the discussion could benefit from some considerations from the authors on possible biological reasons the associations are more or less likely to be real.
R: We agree with the reviewer. The challenge is very complicated in a setting with many confounding variables. We cannot establish from our analysis which is a causal element that determines the progression of the disease and worsens the prognosis. However, we confirmed (with the cox regression analysis that we just added to the results) that NK cells levels were an independent risk factor for worse survival. Our hypothesis was that the reduction of muscular mass lead to reduced levels of IL-15 therefore reducing an important factor for NK proliferation. However, from our results, probably because of the low sample size, we cannot fully confirm this hypothesis. We proved that basal NK levels were reduced in patients with sarcopenia but IL-15 levels were not different. Certainly an impact of IL-15 on NK was proven from the longitudinal results. At the same time, sarcopenic patients had worse survival outcomes. We can therefore conclude that there is an association between low NK, sarcopenia, IL-15 and progression disease, but we cannot state which is the first causal element.We added these considerations in the discussion following your suggestion (see Discussion).
Concerning the longitudinal data, it is not really relevant to identify poor responder's after many months of treatment, it is really the data available at treatment start (when you make an actual choice) that is relevant.
R: We agree with the reviewer. However, we believe that it may be useful to know the change of PBMCs and other inflammatory factors longitudinally to validate the predictive factors found at baseline.
Reviewer 3 Report
This is a report on the biomarker of immune checkpoint inhibitors in advanced NSCLC, and the authors show the amount of circulating NK cells is associated with the effect of ICI.
Major comments.
Although the sample size is small and moreover, the sample size of PD patients on T1 is too small. The result seems meaningful. It may be too long for the follow-up time(T1) of three months to get an early prediction of the benefit of ICI, the majority of PD patients can not continue ICI therapy because of PD earlier than 3 months. Sometimes it is difficult to differentiate PD from pseudo-PD. Early detection of PD is important. A shorter interval time for example two to four weeks may be more useful. It is one of the limitations of this study.
What is the background for NK cell decline? Do more advanced (metastatic) patients have fewer NK cells?
Are there any data on other predictive biomarkers of ICI treatment for example PD-L1, serum Alb, mutation status, or smoking status?
Minor comments.
Line 27. "Natural Killer cells can play an important role in immune escape." What does this sentence mean?
Line 88. 15 -> [15]
Author Response
We would like to thank the Editor, the Associated Editor and the Referees for appreciating our paper and for having raised some interesting points that have allowed us to improve the revised MS.
We believe that now the clarity of the manuscript has been significantly improved. We have been able to address all the comments raised and the detailed point-by-point rebuttal is listed below.
We also updated Tables and figures so as to eliminate supplementary material.
Reviewers' Comments to Author:
Referee 3
This is a report on the biomarker of immune checkpoint inhibitors in advanced NSCLC, and the authors show the amount of circulating NK cells is associated with the effect of ICI.
Major comments.
Although the sample size is small and moreover, the sample size of PD patients on T1 is too small. The result seems meaningful. It may be too long for the follow-up time(T1) of three months to get an early prediction of the benefit of ICI, the majority of PD patients can not continue ICI therapy because of PD earlier than 3 months. Sometimes it is difficult to differentiate PD from pseudo-PD. Early detection of PD is important. A shorter interval time for example two to four weeks may be more useful. It is one of the limitations of this study.
R: Thank you for this very useful comment. We collect blood at 2-3 weeks from the beginning of therapy but of a small number of patients and therefore they were not included in the results. We will certainly keep this in mind for future studies. Anyway we added your consideration in the limit section of the manuscript (see Discussion).
What is the background for NK cell decline? Do more advanced (metastatic) patients have fewer NK cells?
R: We want to thank the author for this observation. Indeed the role of NK cells in human cancers has been proposed for the first time in 1980 revealing higher incidence of cancers in patients with NK cell defects and low NK cells activities in cancer patients (PMID: 35967421). Subsequently, a landmark 11-year following-up study reported a positive correlation between impaired NK cell functions and higher risk to develop numerous types of cancers (PMID: 11117911). Moreover, even Sesti et al. (PMID: 36509928), as already mentioned in our manuscript, reported an immunosuppressed environment in oncologic patients, specifically a low count of cytotoxic NK cells has been described in patients affected by neuroendocrine tumours. We have integrated the discussion as requested.
Are there any data on other predictive biomarkers of ICI treatment for example PD-L1, serum Alb, mutation status, or smoking status?
R: Thank you very much for this observation. We added PD-L1 and smoking status data to Table 1 and to the text accordingly (See Results, 3.1 Characteristic of the Cohort), even if no difference were found at univariate analysis for PD.
Minor comments.
Line 27. "Natural Killer cells can play an important role in immune escape." What does this sentence mean?
We are agree and thank the Reviewer, we have now modified the text as below:
"Natural Killer cells are involved in cancer immune surveillance”
Line 88. 15 -> [15]
Done
Round 2
Reviewer 1 Report
The authors have addressed all suggestions made by this Reviewer by increasing scientific rigor, clarity, and interest for readers. I endorse the publication.